# The Perception among Israeli Gastroenterologists Regarding Treatment of Patients with Biosimilar Medications

**DOI:** 10.3390/medicina59030523

**Published:** 2023-03-07

**Authors:** Vered Richter, Daniel L. Cohen, Anton Bermont, Tzippi Shalem, Efrat Broide, Haim Shirin

**Affiliations:** 1The Gonczarowski Family Institute of Gastroenterology and Liver Disease, Shamir (Assaf Harofeh) Medical Center, Zerifin 70300, Israel; 2The Jecheskiel Sigi Gonczarowski Pediatric Gastroenterology Unit, Shamir (Assaf Harofeh) Medical Center, Zerifin 70300, Israel; 3Sackler School of Medicine, Tel Aviv University, Tel Aviv 39040, Israel

**Keywords:** biosimilar drugs, inflammatory bowel disease, treatment perception

## Abstract

*Background and Objectives*: The ever-expanding entry of biosimilar drugs into the Israeli market requires doctors to decide whether to prescribe these medications. We aimed to assess the prevalence of biosimilar use and Israeli gastroenterologists’ knowledge, experience, and perception of biosimilar treatment. *Materials and Methods*: A cross-sectional survey was conducted among Israeli Gastroenterology Association (IGA) members between March and May 2022 using a structured 20-item questionnaire. *Results*: The questionnaire was completed by 108 gastroenterologists. Sixty-two percent prescribed biosimilars to their patients in the past year. Most of the patients (81%) were biologically naïve and only 19% were switched to a biosimilar. Most gastroenterologists (75%) answered that the effectiveness is the same. The rates of resistance to switching were 19%, 36%, and 70% for patients in remission for over two years, pregnant women, and difficulty reaching remission, respectively. In cases seeing a lack of response after switching, most physicians chose to change the mechanism of action, with only a small percentage returning to the brand-name drug. *Conclusions*: Most Israeli gastroenterologists are not concerned about biosimilars’ safety and efficacy. Despite this, most physicians will prefer the brand-name drug, especially regarding adalimumab. The populations in which physicians most oppose switching are those who have had difficulty achieving remission and pregnant women.

## 1. Introduction

Biosimilars are defined as biological products that are highly similar to and have no clinically meaningful differences from a Food and Drug Administration (FDA)-approved reference product. However, they are not generic or identical because within-product differences may exist.

The FDA approval pathway includes both analytical studies in order to test that the biosimilar is structurally and functionally similar to the reference drug, as well as clinical studies whose purpose is to test whether there are differences in pharmacokinetic, safety, efficacy, and immunogenicity [www.fda.gov/drugs/therapuethic-biologics-application-bia/biosimilars].

Interchangeability requires evidence of bio-similarity plus evidence that switching between the originator and biosimilar does not compromise efficacy and safety.

Biosimilars for IBD have been used in Europe since 2013, the United States since 2016, and Israel since 2016 [1,2]. These medications are used in different therapeutic areas including gastroenterology, inflammatory diseases, oncology, immunology, and hematology. In the United States, the Food and Drug Administration (FDA) has approved 33 biosimilars corresponding to 11 different reference products as of 2021 [3]. The European Medicines Agency (EMA) has approved 86 biosimilar medicines since 2006 [4].

The health system in Israel provides universal healthcare to all citizens. All physicians are able to prescribe biosimilars as they have been approved by the Ministry of Health, and thus are available to all patients through both community medical clinics and hospitals.

However, there is still insufficient evidence and a lack of clear guidelines regarding the preference of one drug over the other and regarding the replacement of the original drug with a biosimilar during treatment. Many questions remain. Is it reasonable, for financial reasons, to switch from the originator to a biosimilar in all patients, or should we exclude specific populations such as those with active disease or pregnant women? How many switches can be made for one patient? Should it be a pharmacy-driven automatic substitution for each prescription, or should it be the prescribing doctor’s decision? If the patient stops taking the biosimilar after switching from the source drug because of worsening of symptoms (secondary loss of response) or side effects, should we reverse switch back to the source drug or try a different biosimilar?

Until enough evidence has accumulated from head-to-head comparisons, and until clear guidelines on the subject emerge, we decided to investigate the perception of Israeli gastroenterologists prescribing these drugs.

The aim of our study was to evaluate Israeli gastroenterologists’ knowledge, experience, and opinions regarding biosimilar medicines, including the decision to start a new biological treatment and switch treatment between an originator and a biosimilar drug.

## 2. Materials and Methods

### 2.1. The Questionnaire

The authors developed an online survey that contained 20 questions regarding the prevalence of biosimilar use, the decision to start a new biological treatment, and switching treatment between originator and biosimilar drugs. The questions were open, multiple-choice, or rating scale questions (range: 0–5). The questionnaire also contained five questions addressing the physician’s personal experience with biologic treatments. Demographic data including gender, age, and job position were also collected (Appendix A).

### 2.2. Data Collection

The questionnaire was sent by e-mail to Israel Gastroenterology Association (IGA) members between March 2022 and May 2022. Kantar, a company that conducts surveys, provided the platform to send emails to the IGA members. The emails to the doctors were sent by the researchers and not by the Kantar company. The payment to Kantar was made by a grant from AbbVie. Apart from this funding, AbbVie had no contact with the physicians and were not involved in collecting the results, processing the statistics, or writing the article.

The study was reviewed and approved by the Shamir Medical Center Institutional Review Board [(Approval No. 0043-22-ASF]). The Ethics Committee did not require informed consent because the questionnaire was answered anonymously and the decision to respond to the questionnaire was considered consent.

### 2.3. Statistical Analyses

Categorical variables were reported as frequency and percentage and continuous variables were reported as median with interquartile range. *T*-test was used to determine a significant difference between groups at a significance level of *p* < 0.05.

## 3. Results

Out of 486 members registered in the IGA, 130 were excluded as they were not practicing gastroenterologists. Thus, the survey was sent to 356 gastroenterologists, of whom 108 completed the questionnaire, giving a response rate of 30.3%. The mean age of the respondents was 47.6 years (±9.48), and 65% were men. Ninety-seven percent had finished their training and 63% had professional experience of more than 10 years. Forty-five percent worked in a hospital only, 16.7% in HMOs, 45.6% in combined hospital and HMOs, and 9% in private practice. The doctors’ workplaces according to geographic regions in Israel are detailed in Figure 1.

Physicians were treating on average 24 IBD patients per month with biologics (range: 1–200). Most of the physicians (67, 62%) have prescribed biosimilar drugs to their patients in the past year. However, the share of biosimilars is still relatively marginal. On average, each physician who prescribed biosimilar drugs in the past year prescribed them to only 8% of their biologically treated patients.

Eighty-one percent of patients for whom biosimilar therapy was prescribed were biologically naïve and only 19% were switched to a biosimilar. Among patients whose prescription drugs were replaced with a biosimilar, 55% were stable at the time of the switch.

When asked, “To what extent do you think a biosimilar drug is effective compared to a brand-name drug?” 75% answered that the effectiveness is the same. Only 13% thought that the original drug is more effective, 2% thought that the biosimilar drug is more effective, and 10% did not know. Among those who have prescribed biosimilars in the past year, 81% answered the same effectiveness and 10% answered that the original drug is more effective, compared with 66% and 17%, respectively, among those who have not prescribed biosimilars in the past year (*p* < 0.05).

When asked about their concern regarding the safety of biosimilar drugs, the average response was 2 on a scale in which 1 represented “not concerned” and 5 represented “concerned very much”.

When initiating a new biologic drug in which there were no binding guidelines, 58% preferred the brand-name drug and 25% preferred a biosimilar with regard to infliximab, while 67% preferred the brand-name drug and 19% preferred a biosimilar in the case of adalimumab (Figure 2).

Regarding switching, 70% of the doctors stated that they would not switch to biosimilar patients who had difficulty reaching remission, 29% would not switch patients with active disease, and 36% would not switch pregnant women (Figure 3).

When asked to answer according to their personal experience, the physicians estimated that 11% of patients lose response after switching to a biosimilar. Moreover, when estimating the increase in the drug immunogenicity after switching, 57% of the physicians answered that none of the patients had an increase in the drug immunogenicity, while 42% answered that a small proportion of patients and only 1% answered that large proportion of patients had an increase in the drug immunogenicity.

In the case of losing response after switching, most physicians (78%) will choose a medication with a different mechanism of action (indicating a degree of agreement 4–5 on a scale of 1–5 regarding the degree of possibility that they will do so, and only 21% will return to the brand-name drug (indicating 4–5 on the same scale regarding the possibility they will return to the brand-name drug).

The patient’s involvement in changing the drug to a biosimilar was rated 4.1 on average, while having the necessary tools (information, time, and so on) to explain to the patient the details of switching was rated 3.3. The physicians rated the statement that there is enough information on biosimilars available from RCT studies as 3.3, and they rated highly (4.0) the desire to learn about biological drugs from reliable sources.

According to the decision of the Israeli Ministry of Health, the brand-name drug can be replaced with a biosimilar “in a stable patient (who is in remission) only, at the discretion of the physician”. [2] When the physicians were asked for their opinion regarding the criteria for what determines a remission that allows switching to a biosimilar, most of them included clinical, laboratory, and endoscopic/imaging remission. However, about a third of them do not require endoscopic/imaging remission for this.

When comparing doctors in the periphery versus those in the center, doctors in the periphery prescribed fewer biosimilar drugs in the past year (13.7% vs. 31.4%, *p* < 0.05), were more concerned about the safety of biosimilars (2.3 vs. 1.9 on a scale of 1–5, *p* < 0.05), assumed that there is a higher percentage of patients with a loss of response (46.7% vs. 23.8%, *p* < 0.05), and felt more uncomfortable switching a patient with a stable disease to a biosimilar (3.5 vs. 2.8 on a scale of 1–5, *p* < 0.05).

## 4. Discussion

The results of our survey show that most gastroenterologists are not concerned about the safety and efficacy of biosimilar drugs. Those who already have prescribed biosimilars in the past year seem to be more confident in their efficacy. However, assuming there are no binding guidelines from health maintenance organizations, most physicians prefer the brand-name drug. This was slightly more common with regards to adalimumab than infliximab.

However, according to the literature, the situation was different until recently. A literature review conducted by Leonard et al. between 2014 and 2017 showed that an overall lack of biosimilar familiarity in United States and European health care settings accompanied concerns about biosimilar safety, efficacy, extrapolation, and interchangeability [5]. A systemic review of 23 studies addressing physicians’ perceptions of the uptake of biosimilars published between 2014 and 2019 showed that 65–67% had concerns regarding these medicines [6]. Our research shows that the situation has changed since then, as the answer to the question “To what extent are you concerned about the safety of the biosimilar” was on average 2 on a 1–5 scale. However, we found that, still, as in those review articles, physicians are more willing to prescribe biosimilars for biologic-naive patients rather than for patients already being treated with biologic medicines. Data analysis according to the distribution to regions showed the same rate of biologics’ use between the periphery and central areas. However, doctors in the periphery were more concerned about the use of biosimilars in terms of safety and efficacy.

Many studies have evaluated the biosimilar CT-P13′s efficacy and safety. The NOR-SWITCH study, conducted in 2014–2015, was a randomized, non-inferiority, double-blind, phase IV trial that showed that switching from infliximab originator to CT-P13 was not inferior to continued treatment with infliximab originator. The frequency of adverse events was similar between the groups [7]. Many other studies conducted since then, including the long-term extension of the NOR-SWITCH study, also showed no difference in the safety and efficacy of the biosimilar CT-P13 [8,9,10,11]. The PROSIT-BIO Cohort evaluated CT-P13′s efficacy, safety, and immunogenicity in patients with IBD. Although there was no direct comparison, the CT-P13 results were consistent with those of infliximab [12]. However, some studies showed a higher risk of clinical relapse, which was not supported by an increase in objective markers, and thus attributed this to the nocebo effect [13]. This is a negative effect of a treatment that is caused by patients’ negative expectations regarding the treatment and that is unrelated to the physiological action of the treatment.

Multiple switchings also appear to be effective and safe according to several studies [14,15]. In a multicenter prospective cohort study, Hanzel J et al. observed similar rates of clinical and biochemical remission at 12 months in patients undergoing multiple switches, a single switch between biosimilars, or a single switch from the originator to CT-P13 [16].

Switching between the originator adalimumab to biosimilars and between adalimumab biosimilars in IBD patients also appear to be safe and effective [17,18]. Tursi et al. compared adalimumab biosimilars (ABP 501, SB5, MSB11022, and GP2017) in IBD treatment in a large IBD Italian outpatient population. There was no significant difference between the four-adalimumab biosimilar regarding effectiveness and safety in both new patients and in patients switching from the adalimumab originator [19].

The biggest resistance to switching in our study was among patients who had a difficult time achieving remission and are now being maintained in remission. Moreover, about a third of the respondents in our study were hesitant to switch to a biosimilar in the case of an active disease. Indeed, in the study of multiple switching conducted by Hanzel J et al., an association was found between remission at the time of switching and subsequent maintenance of remission [16]. Nonetheless, in our study, there was no consensus among physicians regarding biosimilar replacement in stable patients. Regarding pregnant women, 36% of the physicians indicated that they would not switch to a biosimilar. While it is likely that the drug’s mechanism is the same in pregnant women, women have a stronger and more frequent nocebo effect than males, according to reported data [20,21]. The nocebo effect, together with the obstetric consequences that may occur in the event of a disease flare, appear to make physicians hesitant to make a switch in this particular group of patients.

In the case of loss of response to treatment, most of our physicians would choose to change the mechanism of action of the medication, with only a fraction returning to the brand-name drug. It is unclear whether switching patients back (reverse switching) to the formulation of the biosimilar used before would be beneficial in patients who developed adverse events or loss of response after switching. Harmful consequences after switching, as well as their resolution after reverse switching, have been ascribed to both the difference between the formulas of biosimilars and to the nocebo effect.

In a retrospective, multicenter cohort study, which followed patients after switching from the originator infliximab to the biosimilar CT-P13 for at least 52 weeks after the initial switch, reverse switching back to the originator infliximab was observed in 9.9% of cases. Patients with reverse switching were predominantly female (70.7%). Gastrointestinal symptoms (25.5%) and dermatological symptoms (21.8%) were the most reported reasons for reverse switching. An improvement in reported symptoms was seen in 73.3% of patients after reverse switching. Loss of response to CT-P13 was the reason for reverse switching in 12.0% of patients, and 77.8% of them regained response after the reverse switching. The authors concluded that switching back to the originator infliximab seems to be effective in patients who experience adverse effects, worsening gastrointestinal symptoms, or loss of response after switching from the originator infliximab to CT-P13 [22].

The patient’s involvement in changing the drug to a biosimilar is perceived as very important for our physicians, and they feel that they need more tools for doing so in the right manner. Patients’ knowledge about biosimilars, as well as issues around them, was evaluated in a multicenter survey study. Among the responders, 50% worried that biosimilars would be less efficient than the original and 46% were concerned about the safety profile. These findings may explain the nocebo effect [23]. Providing information and working collaboratively with patients are important not only because of patients’ right to be involved in all matters relating to their care, but also because it is the key to minimize the nocebo effect.

The physicians state that there is enough information available from RCT studies, but they would like to learn more about biological treatments from reliable sources. Despite the financial advantages of biosimilars, the success of biosimilar implementation depends on well-informed physicians explaining the safety and efficacy of these drugs with their patients.

The strength of this study is that a range of gastroenterologists from across Israel participated. However, there are some limitations. First, the researchers formulated the questionnaire, and its validity is unclear. Second, we do not know why doctors chose not to participate.

## 5. Conclusions

Most Israeli gastroenterologists are not concerned about biosimilars’ safety and efficacy. Despite this, most physicians’ preference will be the brand-name drug, especially when it comes to adalimumab. The populations in which physicians most commonly oppose switching to biosimilar treatment are those who have had a hard time achieving remission and pregnant women. There is no consensus regarding biosimilar replacement in stable patients. While data are available, the physicians stated they would like to learn more about biological treatments from reliable sources. With the rapidly gained momentum of biosimilars and the expected introduction of more biosimilars to the IBD field in the coming years, studies such as this one are very valuable. Its utility is in both directing the coming studies to answer unsolved questions and in guiding the delivery of special educational knowledge to IBD practicing physicians to relieve their concerns.

## Figures and Tables

**Figure 1 medicina-59-00523-f001:**
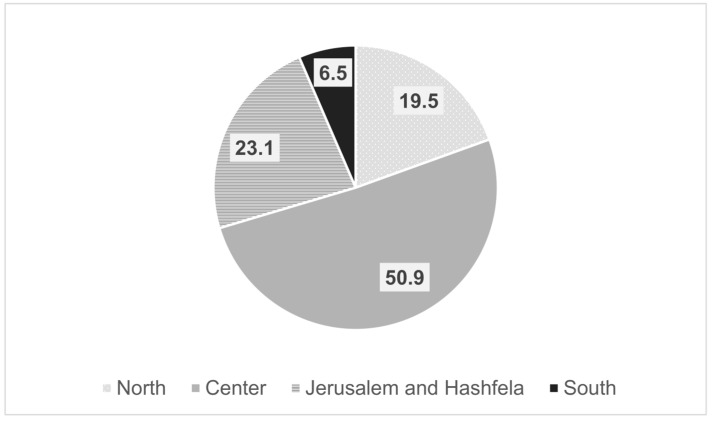
Geographic location of physicians’ workplace.

**Figure 2 medicina-59-00523-f002:**
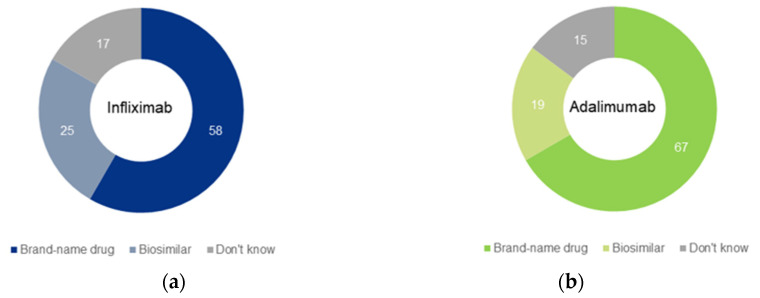
Treatment preference in the case of no binding guidelines, regarding infliximab (**a**) and adalimumab (**b**).

**Figure 3 medicina-59-00523-f003:**
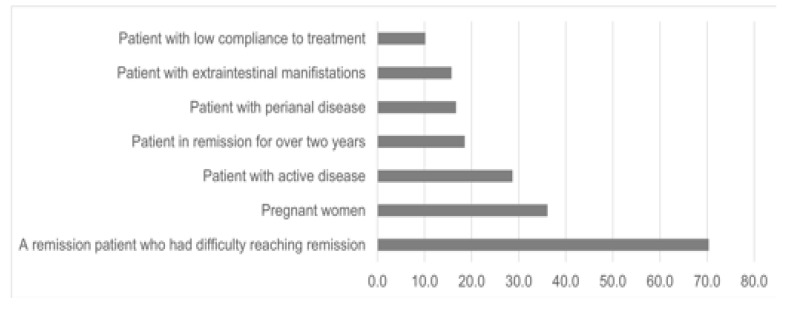
Percentage of the physicians who resist biosimilar switching in different patients’ populations.

## Data Availability

The data presented in this study are available upon reasonable request submitted to the corresponding author. The data are not publicly available owing to privacy.

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
