# Peer review of "The Perception among Israeli Gastroenterologists Regarding Treatment of Patients with Biosimilar Medications"

_medicina, 2023, doi:10.3390/medicina59030523_

Round 1

Reviewer 1 Report

This study is of high importance, well designed and written, and contains a clear conclusion. with the rapidly gained momentum of biosimilars and the expected introduction of more biosimilars to the IBD field in the coming years such studies are very valuable. It's utility in both directing the coming studies to answer the unsolved questions and to guide delivering special educational knowledge to IBD practising physicians to relieve their concerns. [ I would suggest adding this to the conclusion if you agree].

I find this study suitable for publication. Though I have minor comments:

In the survey, question 4, it is not clear enough what a doctor might choose if he switches Remicade (IFX originator) to adalimumab biosimilar. I assume in this case, this patient would be considered "biologic naive" in the survey although he is Adalimamb naive only and not biologic naive.  could you make this clear?

Other studies that you may find worth citing:

Tursi, Antonio et al. “Comparison of Performances of Adalimumab Biosimilars SB5, APB501, GP2017, and MSB11022 in Treating Patients with Inflammatory Bowel Diseases: A Real-Life, Multicenter, Observational Study.” Inflammatory bowel diseases, izac092. 17 May. 2022, doi:10.1093/ibd/izac092

  1. Fiorino G, Manetti N, Armuzzi A, et al. The PROSIT-BIO cohort: a prospective observational study of patients with inflammatory bowel disease treated with infliximab biosimilar. Inflamm Bowel Dis. 2017;23:233–3.

Peyrin-Biroulet, Laurent et al. “Changes in inflammatory bowel disease patients' perspectives on biosimilars: A follow-up survey.” United European gastroenterology journal vol. 7,10 (2019): 1345-1352. doi:10.1177/2050640619883704

Author Response

Dear Reviewer

Thank you for the time to review our study.

The changes we made are attached.

Reviewer 2 Report

The authors questioned members of the Israeli Gastroenterology Association and collected questionnaires from 108/356 to evaluate the perception of biosimilars in Israel. However, this raises several concerns, as shown below, and it is unclear whether the results will be useful for addressing future issues worldwide.

-Present the acceptance of biosimilars in each country in the INTRODUCTION.

-What is the population of these participating IGA members? I am not sure how many physicians in Israel who may use biosimilars for IBD are covered.

-I don't see the significance of why you presented data on participants by region in Israel, so the authors should either remove it or discuss it based on the results.

-What percentage of participants use Biosimilars? How many of their hospitals have access to the biosimilar, and how many of them are reluctant to use it simply because they belong to hospitals that do not have access to a biosimilar for adalimumab?

-How many of the participants were aware of the results of the Key study; if they were aware of the results of NOR-SWITCH and many other studies, this result would be very different.

Author Response

(The authors gave the same response as above.)
